# I love the way you love me: Responding to partner's love language preferences boosts satisfaction in romantic heterosexual couples

Olha Mostova[1], Maciej Stolarski[2]*, Gerald Matthews[3]

1 Doctoral School of Social Sciences, University of Warsaw, Warsaw, Poland, 2 Faculty of Psychology, University of Warsaw, Warsaw, Poland, 3 Department of Psychology, George Mason University, Orlando, FL, United States of America

* mstolarski@psych.uw.edu.pl

## Abstract

Chapman's Love Languages hypothesis claims that (1) people vary in the ways they prefer to receive and express affection and (2) romantic partners who communicate their feelings congruent with their partner's preferences experience greater relationship quality. The author proposes five distinct preferences and tendencies for expressing love, including: Acts of Service, Physical Touch, Words of Affirmation, Quality Time and Gifts. In the present study partners ($N$ = 100 heterosexual couples) completed measures assessing their preferences and behavioral tendencies for a) expressions of love and b) reception of signs of affection, for each of the five proposed "love languages". Relationship satisfaction, sexual satisfaction and empathy were also assessed. The degree of the within-couple mismatch was calculated separately for each individual based on the discrepancies between the person's felt (preferred) and their partner's expressed love language. The joint mismatch indicator was a sum of discrepancies across the five love languages. Matching on love languages was associated with both relationship and sexual satisfaction. In particular, people who expressed their affection in the way their partners preferred to receive it, experienced greater satisfaction with their relationships and were more sexually satisfied compared to those who met their partner's needs to lesser extent. Empathy was expected to be a critical factor for better understanding of and responding to the partner's needs. Results provided some support for this hypothesis among male but not female participants.

## Introduction

Everyone has their own preferences for expressing and receiving romantic feelings. Based on his clinical observations Chapman [1] identified five distinct ways in which people show and want to receive the signs of commitment. He labeled them *'love languages'* (LLs) and proposed division into the following domains: 1) words of affirmation (verbal compliments, or words of appreciation), 2) quality time (time spent together implying focused attention of both partners), 3) receiving gifts (visual symbols of affection), 4) acts of service (helping mate with necessary tasks) and 5) physical touch (from holding hands to sexual intercourse).

**Data Availability Statement:** All relevant data are within the paper and its Supporting Information files

**Funding:** This work was supported by the University of Warsaw, from the funds awarded by the Ministry of Science and Higher Education in the form of a subsidy for the maintenance and development of research potential.

**Competing interests:** The authors have declared that no competing interests exist.

Chapman [1] used a metaphor of a 'love tank', which reflected people's emotional need to feel loved. The 'love tank' of both partners is filled when each of them expresses the affection in a way another one prefers to receive it. According to Chapman and Southern, "once you identify and learn to speak your spouse's primary love language. . .you will have discovered the key to a long-lasting, loving marriage" [2, p. 18]. Thus, the author is convinced that conducting interventions designed to foster better understanding of the partner's LLs, as well as educating people on how to put this knowledge into practice would lead to a greater relationship mainte-nance and satisfaction.

People often speak and understand their primary LLs, but they may "learn a secondary love language" [2, p.14]. For example, the husband may be aware of his wife's desire to receive com-pliments–words of affirmation, although he himself prefers being physically touched. There-fore, LLs can be divided into those in which people tend to communicate to their partners (expressed love language) and those they prefer to receive in order to keep their emotional 'love tank' full (felt love language).

The assumption of basic and fundamental need for love and affection is well-established and empirically supported [3–5]. Multiple research findings also support the notion that human's need for love and affection boosts both personal well-being [e.g. 6] and satisfaction in various types of relationships [e.g. 7, 8].

Research has also addressed the effects of similarity between partners on relationship out-comes. It is well established that similarity and convergence of different types in romantic part-ners promotes greater relationship satisfaction. For example, similarity in communication values, such as ego support or conflict management, has been shown to promote attraction and greater relationship satisfaction in couples which described themselves as 'seriously involved' [9].

Gonzaga and colleagues [10] also found that similarity and convergence in the personality between partners promotes similarity in their shared emotional experiences and relationship quality. Yet another study demonstrated that romantic couples who converge in their emo-tional experiences manifest greater relationship cohesion, while their relationships are less likely to dissolve [11]. In another study perceived similarity in text messaging, including fre-quency of initiating a text message exchange and expressing affection, was associated with greater relationship satisfaction [12].

Sexual satisfaction is another broadly-studied concept in the research on couples and romantic relationships. Various studies found an association between sexual satisfaction and relationship satisfaction [13, 14]. For instance, in a study conducted on 387 couples, sexual sat-isfaction and communication independently predicted the relationship satisfaction [14]. Thus, romantic partners who are having difficulties communicating, but are at the same time sexu-ally satisfied, will experience greater marital satisfaction. Similarity in personality traits between marital partners also proved to predict sexual satisfaction [15] In another study, de Jong and Reis [16] found that complementarity, but not similarity, in sexual preferences pre-dicted sexual satisfaction, consistent with LL hypothesis. Being aware of one's partner's prefer-ences and acting accordingly may increase the couple's sexual satisfaction.

The act of giving may potentially be more satisfying than benefiting oneself. One study [17] found that spending more of one's income on others predicts greater happiness than spending money on themselves. Prosocial spending is positively correlated with greater happiness in both poor and rich countries and even recalling the past instance of spending money on others has a causal impact on happiness [18]. Thus, it is possible that helping others to match their needs produces a greater emotional benefit than receiving and caring for oneself. Thus, com-patibility in LLs may benefit the person as a giver, as well as a receiver.

Empathy is often believed to play a crucial role in relationships of various kinds and to be a key component of effective communication. It can be defined as the "reactions of one individual to the observed experiences of another" [19, p.113]. Empathy is primarily related to one's ability to understand and share the emotional experiences of others [e.g., 20]. According to Davis [19, 21], empathy can be divided into four domains, including: empathic concern, personal distress, perspective taking and fantasy.

Research demonstrates a positive relation between a partner's empathic accuracy and their degree of relationship satisfaction [22, 23], when it is present in mundane and nonconflictual settings. Various studies have also found that people with higher levels of empathy are better in detecting their partners' needs and providing them with higher-quality advice and instrumental support [24–26]. On the other hand, subjective perceptions of empathy may be more important for relationship satisfaction than empathic accuracy [27], i.e., the objective ability to determine one's partners views.

Empathic accuracy implies accurate perception of people's mental states, including thoughts and feelings [24] people that are more proficient in mentalizing may also be more skillful in understanding their partner's needs and preferences. Such an awareness may in turn increase the likelihood of fulfilling such needs and make them choose and adjust their behavior more consciously. Thus, empathy may support congruence in LLs. Although there were some attempts to examine the factors that may moderate the relationship satisfaction when partners are misaligned in their LLs [e.g. 28], to our knowledge, none of the previous studies have examined the potential mediators that may drive LL matching, leading to elevated satisfaction. An empathic individual may be both more effective in giving the form of love desired by the partner, and in guiding the partner towards understanding their own needs.

Despite the great popularity that Chapman's work had gained worldwide among both clinicians and general public, the concept of LLs remains relatively unstudied. Egbert and Polk [29] developed the Love Languages Scale based on concepts found in Chapman's [1] LLs and suggested that the five-factor LL model had some psychometric validity. A confirmatory factor analysis demonstrated significant relationship between the five LLs and Stafford, Dainton, and Haas' [30] relational maintenance typology, thus supporting their construct validity.

In one version of the scale participants responded about how they tend to feel love. To remain consistent with the literature we followed the term 'felt LL' that was used in Egbert and Polk's [31] study to infer the preferred way of receiving love from one's partner. However, it should be noted that two forms of the LL scale provide the data on what makes people feel loved without directly examining what is being preferred, which might be a potential conceptual problem with the LL measure.

Another study utilized measures of the autonomic nervous system, such as skin conductance and respiration rate [32]. First, 89 participants were asked to complete Chapman's LLs questionnaire. Next, their psychophysiological responses (skin conductance, heart rate and pulse rate) were measured, while they were listening to the recorded imagery scripts (imaginal exposure and guided imagery) describing each of the LLs. It was found that participant' arousal level increased when they were listening to their dominating or preferred LL. Specifically, a significant association was found between hearing the imagery script of their primary LL and person's heart rate and skin conductance, although no significant increase was found in the respiration rate. In addition, good internal consistency and reliability was reported for the scale measurement of the preferred LL [33].

Other aspects of Chapman's [1] claim have not been supported. Egbert and Polk [31] tested 84 university student couples, who had been together for at least two months. The aims was to check whether partners well-matched in their reported preferences and expressions of LLs reported greater relationship quality. They were grouped into three categories including

matched, mismatched and partially matched; however, no significant effect of matching on relationship quality was found. According to later research [28] sharing the same primary (expressed) LL again did not result in relationship satisfaction. However, the researchers found that female participants' self-regulation significantly improved partners' satisfaction when their LLs were misaligned.

Veale [34] aimed to test whether becoming aware of one's partner's preferred LL would result in behavioral adaptation in the absence of any identified external motivation. First, the researcher checked whether participants felt that the love expression used by their partner was correctly identified. Second, descriptions of the LL profiles and category membership were provided to each participant, and the couples were given a brief overview of the love expression behavior related to each of five categories. Additionally, the study considered whether LL expression knowledge would influence participants' emotional state and behavior, as well as whether efforts to make behavioral accommodation would be noticed by their partners. The research found no statistically significant difference between pre- and posttest means for any of the categories, although in general respondents agreed that their LL was correctly identified. It should be noted that LLs in the study were evaluated categorically using Chapman's Love Language Profile, which shares the assumption that people have one dominating LL that they both feel and express the most.

## The present study: Aims and hypotheses

Chapman's [1] basic claim is that people who "talk" their partner's LL are more satisfied in their relationships, but empirical support for the LL hypothesis remains equivocal. In the present study, we aimed to extend previous findings [e.g. 31] by adopting a novel, continuous approach to measure match/mismatch between partners' LLs. Treating LL mismatch as a continuous dimension may preserve information on level of mismatch that is lost by assessing it on a categorical basis, and provide a more sensitive test of its relationships with other variables. First, we aimed to check whether romantic partners who receive affection in ways consistent with their partner's style of expressing love report higher relationship satisfaction and sexual satisfaction. We asked people separately about when they feel loved the most (i.e., preference for receiving love) and how they prefer to express affirmation of love. The aim was to assess the partner's match not only in term of their dominant love language, but to also take into account possible that one's preferences for receiving love may differ from one's style of expressing and communicating love.

Second, we aimed to distinguish "actor" effects (mismatch lowers one's own satisfaction) and "partner" effects (mismatch lowers the other person's satisfaction). Chapman's [1] hypothesis emphasizes the former, but there may also be a reciprocal effect, i.e., if my partner feels s/ he is not receiving the love affirmations s/he wants, the partner's deficit in feeling loved may lower my satisfaction. This approach provided a full picture of the interplay between both partners' expressed and felt LLs, contrary to Chapman's assumption that people give and feel only one dominating LL.

We aimed to test four hypotheses:

1. Relationship satisfaction and sexual satisfaction are negatively associated with the mismatch between the person's felt and their partner's expressed LLs (actor effect). This is the central claim of Chapman [1]. His account of the LLs hypothesis does not imply that the importance of LLs varies for men and women, but we tested for possible gender differences on an exploratory basis.

2. Relationship satisfaction and sexual satisfaction are negatively associated with the mismatch between the person's expressed and their partner's felt LLs (partner effect). The partner's feelings of not being adequately loved may influence actor satisfaction.

3. Matching in LLs is associated with empathy levels, given that empathy is likely to support adaptation to the partner's needs. Given that empathy is a multifaceted construct, we hypothesized that its most adaptive facets, i.e., empathic concern (emotional empathy) and perspective taking (cognitive empathy), would be the aspects related to LLs.

4. Empathy is associated with higher relationship and sexual satisfaction, as in multiple previous studies [e.g. 35–37]. We also anticipated testing the mediating role of empathy in the LL mismatch–satisfaction association, contingent on support for the third and fourth hypotheses.

## Methods

### Participants

The working sample consisted of 100 heterosexual couples (100 men, 100 women), who were sexually active with their partners. All the participants were in a current romantic relationship for at least 6 months. Relationship length varied between 6 months and 24 years ($M = 3.5$ years). The initial sample consisted of N = 110, but data from 10 couples was discarded, as only one partner have completed the questionnaire or there was a significant amount of incomplete or missing information for one or both partners. When an individual item was not scored by the respondent, the average over available items was calculated and multiplyed with the number of items in the questionnaire to replace the missing value.

Age of the participants ranged from 17 to 58 (mean: 27.34). Among male participants age varied from 18 to 58 with a mean value of 28.58 ($SD = 9.68$). Female age ranged from 17 to 57 with an average value of 25.10 ($SD = 7.89$).

The sample was culturally diverse and it included representatives of 31 nationalities. The most commonly occurring nationalities were Ukrainians (N = 67), Poles (N = 24), Belgians (N = 15), Russians (N = 13), Americans (N = 12) and Swedes (N = 12). Majority of both male and female participants indicated their marital status as never married ($n = 116$), followed by married ($n = 63$), "other" ($n = 13$) and divorced ($n = 8$).

The participants were recruited using social media and personal connections. The online questionnaire was distributed to volunteers that provided written consent and reported being in the romantic relationships with their current partner for at least 6 months and were either native speakers of English and/or communicated in English to each other, or reported having sufficient fluency to freely and effortlessly communicate with native speakers (i.e., at least B2 level). The data were collected between October 2018 and March 2019. The subjects spent an average of 23 minutes responding to the questionnaire and were not rewarded. The attention check items with reverse wording were used respectively for each inventory to prevent incomplete and low-effort answers.

All participants gave their informed consent to participate in the study of "communication and satisfaction among couples" that is intended to gather information about various aspects of romantic relationships. They were also informed that their answers will not be assessed individually and that they will not receive any feedback on their results as a couple). We asked participants about the relationship's length and the time since the first sexual intercourse with a current partner to ensure that the participants were sexually active with their current partners, as well as to eliminate the possibility of the 'honeymoon effect' [38]. We also asked both

partners about their fluency in English language. Couples were asked not to discuss or compare their results with each other.

## Materials and procedure

LLs were assessed using two methods. The first was the forced-choice method used by Egbert and Polk [31], while the second one included two forms of a LL scale (LLS) requiring independent ratings of preferences, which was developed and validated by Egbert and Polk [29].

First, participant received a forced-choice LL measure [31] including the following instructions: "I feel the most loved when my partner: (1) physically touches me (i.e., gives a hug, gives a kiss, holds my hand, touches me), (2) helps me out (i.e., running an errand, finishing a chore for me, helping me out, helping to keep things cleaned up), (3) spends quality time with me (i.e., really listening, doing something we both like, engages in quality conversation, spending free time), (4) says encouraging words (i.e., compliments, expresses appreciation for me, gives me credit for something I did, gives me positive comments), and (5) gives me gifts (i.e., a thoughtful birthday gift, a greeting card, a present for no special reason, a gift after being away)". This item served only for the purpose of its later comparison to the two versions of LLS in order to check for the consistency of participant's responses. Thus, this tool was not used as the main indicator of couples' LLs in the subsequent analysis.

The LLS assessment [29] involved four behavioral indicators representing each of the ways to feel and express affection. In the first form of the LLS participants were asked to rate the extent to which they tend to express (or *give*) love to their partners by engaging in the listed behaviors. The second form of the LLS included the same list of behaviors, but this time participants were asked to rate the extent to which each expression makes them *feel* loved by their partners. Thus, each version of LLS consisted of 20 self-report items, including four items per each of the five LLs described by Chapman [1], e.g., "I tend to express my feeling to my partner by telling that I appreciate him/her". The items were rated using a five-point Likert-type scale to assess the extent of agreement on a scale of 1 (*strongly disagree*) to 5 (*strongly agree*). Egbert and Polk [29] reported sufficient construct validity and reliability of the questionnaire in assessing LLs.

As both partners responded to the two versions of LLS, we obtained information about how they tend to express and prefer to feel love. We could then calculate the discrepancy between these measurements. Thus, the degree to which one's (partner) preferred to feel love (e.g., when holding hands), as rated on the Likert scale, differed from the second partner's (actor) degree of expressing love in this way (e.g., by holding hands) served as an indicator of extent to which the respondents felt that their partners were meeting needs in terms of LLs. The sum of the discrepancy scores on the five individual LL components then reflected the degree of match or mismatch.

To remain consistent with the literature we used the term 'felt LL' to infer the preferred way of receiving love from one's partner, as we were following the term used by in the previous studies. However, it should be noted that LLS measure provides the data on what makes people feel loved without directly examining what is being preferred, which might be a potential conceptual problem with the LL measure proposed by Egbert and Polk [29].

In particular, a value of 0 indicated a complete match between the way the respondent preferred to feel and how their partner expressed love (e.g., participant X rates preference for being complimented as 5, and X's partner rates his/her expression of compliments as 5). A negative value meant that one's partner does not express a form of love to the extent desired (e.g. participant X rates preference for being complimented as 5, and X's partner rates his/her expression of compliments as 1). A positive value indicated that one's partner overly expressed

a form of love that the participant did not require to such an extent (e.g. participant X rates preference for being complimented as 1, and X's partner rates his/her expression of compliments as 5). Because our interest was in overall mismatch, we converted all discrepancy scores to absolute values and computed the item-level sum of the four scores for each LL. We then summed these discrepancy scores for each individual, to provide an overall index of mismatch between one's preferences for being loved and partner's ways of expressing love, based on the 20 item-level discrepancy scores. Our primary outcome measure was thus overall LL mismatch; higher scores indicated greater mismatch. The internal consistency of the calculated sum of discrepancies in our study amounted to Cronbach's $\alpha = .61$ for male and $\alpha = .70$ for female participants.

In addition, the internal consistency was confirmed to be acceptable or good for all of the five LLs scales before they were combined to create the discrepancy scores. In particular, the results suggested a good fit for each of the five expressed LLs: Acts of Service ($\alpha = .70$), Physical Touch ($\alpha = .89$), Words of Affirmation ($\alpha = .79$), Quality Time ($\alpha = .78$) and Gifts ($\alpha = .77$). The internal consistency was also confirmed for the five felt LLs scales: Acts of service ($\alpha = .74$), Physical Touch ($\alpha = .85$), Words of Affirmation ($\alpha = .82$), Quality Time ($\alpha = .75$) and Gifts ($\alpha = .84$).

Participants' sexual satisfaction was assessed using the Index of Sexual Satisfaction questionnaire [ISS, 39]. The scale consists of 25-Likert-type items (e.g., "I enjoy the sex techniques that my partner likes or uses", or "I feel that my sex life is boring") and demonstrated high internal consistency among men ($\alpha = .91$) and women ($\alpha = .92$). The participants rated the statements on a 7-point Likert-type, where 1 indicated *none of the time* and 7 indicated *all of the time*. The items that implied lower sexual satisfaction (e.g., "I feel that my sex life is boring") were reversed scored and the total score was calculated as a sum of all item scores. Higher scores indicated greater level of sexual satisfaction.

Relationship satisfaction was measured using the Relationship Assessment Scale [RAS, 40], one of the most frequently used questionnaires used for studying relationship quality. RAS consists of seven items measuring general relationship satisfaction (e.g., "To what extent has your relationship met your original expectations?"). Respondents are asked to rate each statement using a 5-point scale. High internal consistency of the measure was confirmed for male ($\alpha = .85$) and female ($\alpha = .87$) participants.

Interpersonal Reactivity Index [IRI; 19] was applied as a multi-dimensional assessment of empathy. This widely used self-report metric comprises 28 items. The participants are asked to rate their response on a Likert-type scale ranging from 1 ("*does not describe me well*") to 5 ("*describes me very well*"). IRI consists of four distinct subscales, including: 1) perspective taking–the tendency to spontaneously adopt the psychological point of view of others; 2) fantasy–indicating tendencies to transpose oneself imaginatively into the feelings and actions of fictitious characters in books, movies, and plays; 3) empathic concern–assessing other-oriented feelings of sympathy and concern for unfortunate others; and 4) personal distress–measuring self-oriented feelings of personal anxiety and unease in tense interpersonal settings.

We hypothesized that perspective taking and empathic concern scales would be negatively associated with LL mismatch, and positively related to satisfaction measures. Fantasy did not appear directly relevant to LLs, whereas personal distress might be positively associated with mismatch. Internal consistencies were acceptable for all four subscales among male subjects, including perspective taking ($\alpha = .74$), fantasy ($\alpha = .77$), empathic concern ($\alpha = .70$), and personal distress ($\alpha = .70$). In female participants the indicators were acceptable for the empathic concern ($\alpha = .80$) and fantasy ($\alpha = .75$) subscales, but poorer for perspective taking ($\alpha = .62$) and personal distress ($\alpha = .60$).

# Results

## Preliminary analyses

This section reports descriptive statistics for the study variables, as well as tests for gender differences. We also tested within-couple correlations to investigate "assortative mating"; the extent to which people partner with those of similar characteristics to themselves.

Table 1 shows LL preferences on the forced-choice measure. Quality time was the most frequently declared LL, followed by physical touch, acts of service, words of affirmation and receiving gifts. In accordance with prior research [31], there was no significant association between gender and participants' responses in the forced-choice LL measurement, $\chi^2_{(1)} = 14.85$, p = .25 (see Table 1).

Table 2 shows means and SDs for the continuous scores. The rank-ordering of means differed a little from the forced-choice data, in that Physical Touch was the highest-rated preference, in both genders, although Quality Time was also highly rated. Also, contrasting with the forced-choice data, women obtained higher mean scores in four out of five LLs dimensions–all except for acts of service. It seems that women generally preferred to receive love from their partners more intensely than men did (see Table 2). Table 2 also shows data for ratings of expressed love. The order of preferences was similar to that for feeling preferences. Participants tended to express love to their partners primarily by physically touching them, followed by spending quality time together, saying words of affirmation, doing acts of service and giving gifts. No significant sex difference was observed for any but one expressed LL: female participants scored slightly higher than male participants in the expression of quality time LL.

Table 2 shows that, despite the gender differences in felt LLs, men and women did not differ on the overall indicator of LL mismatch, i.e., inconsistency between one's LL preferences and the partner's ways of expressing love. The mismatch indicator does not capture directional biases; i.e., whether mismatch results from the partner providing a deficiency of acts of love, or providing more than the person wants. The raw differences were calculated by subtracting the partner's expressed LL from the actor's felt (preferred) level of LLs. Positive values indicate deficiency, while negative values signify excess. The analysis demonstrated that the female participants generally experienced a "lack" in terms of their preferred levels of receiving LLs, particularly in the case of Quality Time. On the other hand, male participants seemed to perceive or feel the affection to a lesser extent than their female partners reported expressing it, thus indicating an "excess". However, this was not the case for men's Acts of Service LL.

Table 2 also shows the cross-gender correlations for LLs. Overall LL mismatch was quite substantially correlated across the couples. However, at the level of individual feelings and expressions, only two out of five LLs–acts of service and gifts–showed significant correlations (assortative mating). Thus, the least valued LLs showed assortative mating but the most valued did not, for both feelings and expressions.

Table 2 also provides descriptive statistics for the satisfaction and empathy variables. There were no significant gender differences in satisfaction, but consistent with other

**Table 1. Results of chi-square test and descriptive statistics for gender differences.**

|  | Physical touch | Acts of Service | Quality Time | Words of Affirmation | Receiving Gifts |
|---|---|---|---|---|---|
| Men | 28 (28%) | 14 (14%) | 42 (42%) | 16 (16%) | 0 (0%) |
| Women | 30 (30%) | 14 (14%) | 41 (41%) | 11 (11%) | 4 (4%) |

*Note.* $\chi^2_{(1)}$ = 9.52, p = .25. Numbers in parentheses indicate column percentages.

**Table 2. Descriptive statistics, between-group mean comparison, and Pearson's correlation coefficient N = 100 couples.**

|  | Women | | Men | |  |  |  |
| --- | --- | --- | --- | --- | --- | --- | --- |
|  | **M** | **SD** | **M** | **SD** | **T** | **G** | **r** |
| Relationship Satisfaction | 4.22 | 0.67 | 4.26 | 0.65 | .52 | −0.06 | .46** |
| Sexual Satisfaction | 5.78 | 0.82 | 5.77 | 0.80 | .12 | 0.01 | .43** |
| E Acts of Service | 3.80 | 0.75 | 3.90 | 0.72 | 1.16 | −0.14 | .24** |
| E Physical Touch | 4.66 | 0.66 | 4.50 | 0.64 | −1.68 | 0.25 | −.01 |
| E Words of Affirmation | 4.31 | 0.62 | 4.28 | 0.71 | −.34 | 0.04 | .11 |
| E Quality Time | 4.50 | 0.55 | 4.31 | 0.73 | −2.04* | 0.29 | −.04 |
| E Gifts | 3.57 | 0.91 | 3.54 | 0.91 | .29 | 0.03 | .34** |
| F Acts of Service | 3.99 | 0.74 | 3.86 | 0.78 | −1.38 | 0.17 | .22* |
| F Physical Touch | 4.67 | 0.56 | 4.44 | 0.67 | −2.77* | 0.38 | .10 |
| F Words of Affirmation | 4.46 | 0.63 | 4.15 | 0.73 | −3.32* | 0.47 | .09 |
| F Quality Time | 4.62 | 0.45 | 4.30 | 0.66 | −4.15* | 0.56 | .01 |
| F Gifts | 3.62 | 0.95 | 3.31 | 1.07 | −2.50* | 0.30 | .26** |
| LL Mismatch | 3.32 | 2.13 | 3.61 | 1.95 | 1.43 | 0.14 | .50** |
| Perspective Taking | 3.84 | 0.59 | 3.59 | 0.70 | −2.68** | 0.39 | −.05 |
| Fantasy | 3.71 | 0.76 | 3.23 | 0.80 | −4.85** | 0.62 | .17 |
| Empathic Concern | 3.89 | 0.71 | 3.31 | 0.67 | −6.66** | 0.84 | .22* |
| Personal distress | 2.95 | 0.61 | 2.46 | 0.68 | −5.70** | 0.76 | .01 |

*Note.* E = expressed; F = felt; LLs = love languages. LL Mismatch reflects the misfit between one's felt and their partner's expressed love languages accumulated for all LLs; higher values indicate poorer fit. Hedges' g is an effect size indicator endorsed for paired samples t-test (see King & Minium, 2003)

*p < .05,

**p < .01.

studies [e.g. 41] female participants' advantage in empathy was observed across all the IRI dimensions. Of the IRI scales, only empathic concern showed a significant though small-magnitude correlation.

## Hypotheses testing

In line with our first hypothesis, we obtained a significant negative association between relationship satisfaction and the LL mismatch indicator, in both men and women (see Table 3). Thus, the greater the discrepancy between preferred and felt LLs, the less satisfied the participants were with their relationships. Similar associations were observed for sexual satisfaction. The actor effects of LL mismatch were stronger in men, particularly for sexual satisfaction, for which the correlations were -.37 (men) and -.21 (women).

Discrepancies in three LLs appeared to be particularly important for participants' relationship and sexual satisfaction (see S1 Appendix). Specifically, mismatch in Physical Touch, Words of Affirmation and Quality Time LLs separately were significantly associated with both male and female partners' relationship satisfaction. Each of these love languages separately was also significantly associated with men's sexual satisfaction. However, among female participants this association was only significant for the Physical Touch and Quality Time LLs, and their sexual satisfaction.

In addition, supporting the second hypothesis, partner effects proved significant–the respondent's satisfaction with their relationship was associated with their partner's LL mismatch. Partner effects were significant for both men and women. It may be as important to properly respond to the partner's LL needs as to have one's own LL preferences satisfied.

**Table 3. Bivariate correlations and internal consistencies ($\alpha$) for scales included in the present study.**

| | 1. | 2. | 3. | 4. | 5. | 6. | 7. | 8. | 9. | 10. | 11. | 12. | 13. | 14. |
|---|---|---|---|---|---|---|---|---|---|---|---|---|---|---|
| Men | | | | | | | | | | | | | | |
| 1. LL Mismatch | --- | | | | | | | | | | | | | |
| 2. Relationship satisfaction | -.36** | (.85) | | | | | | | | | | | | |
| 3. Sexual satisfaction | -.37** | .67** | (.91) | | | | | | | | | | | |
| 4. Empathic concern | -.27** | -.02 | .14 | (.70) | | | | | | | | | | |
| 5. Perspective taking | -.34** | .17 | .29** | .46** | (.74) | | | | | | | | | |
| 6. Fantasy | -.28** | -.02 | .07 | .46*** | .34** | (.77) | | | | | | | | |
| 7. Personal distress | -.18 | -.09 | -.09 | .27** | .04 | .30** | (.70) | | | | | | | |
| Women | | | | | | | | | | | | | | |
| 8. LL Mismatch | **.50**** | -.40** | -.40** | -.16 | -.26** | -.29** | -.12 | --- | | | | | | |
| 9. Relationship satisfaction | -.31** | **.46**** | .34** | .08 | .19 | .13 | .00 | -.30** | (.87) | | | | | |
| 10. Sexual satisfaction | -.25* | .32** | **.43**** | -.03 | .08 | .00 | -.07 | -.21** | .55** | (.92) | | | | |
| 11. Empathic concern | -.17 | -.16 | .00 | **.22*** | .01 | .07 | .09 | -.07 | .02 | .18 | (.80) | | | |
| 12. Perspective taking | -.12 | -.05 | -.10 | .02 | -.05 | .12 | .13 | -.05 | -.06 | -.02 | .32** | (.62) | | |
| 13. Fantasy | -.10 | -.13 | -.07 | .23* | .15 | **.17** | .29** | -.02 | .19 | .19 | .40** | .13 | (.75) | |
| 14. Personal distress | .03 | -.20* | -.06 | .02 | -.07 | -.01 | .10 | -.23* | -.24* | -.13 | .16 | -.10 | .13 | (.60) |

*Note*. Partner effects are shadowed in light grey. Assortative mating effects are provided in bold font.

The third hypothesis was that cognitive and emotional facets of empathy would be associated with lower LL mismatch (see Table 3). The results indicated some significant actor effects in men. In particular, empathic concern ($r$ = -.27, $p$ < .01) and perspective taking ($r$ = -.34, $p$ < .01) components of empathy were higher in male individuals with smaller LL discrepancy, consistent with the hypothesis. However, this was not the case for female participants (see Table 3). Women whose partners scored higher on perspective taking ($r$ = -.26, $p$ < .01) and fantasy dimensions of IRI also showed lower LL mismatch. Our hypothesis regarding the links between empathy and LL matching was then only partially supported.

Contrary to the fourth hypothesis, IRI empathy dimensions proved generally unrelated to both relationship satisfaction and sexual satisfaction. The only two exceptions referred to men's perspective taking (positive actor effect on sexual satisfaction in men) and personal distress (women's distress was related to lower men's relationship satisfaction). Although some scholars [e.g. 42] would emphasize that the absence of the association is not disqualifying the possibly of mediation, we found that in the present analysis there was no reason to test models in which empathy mediated effects of LL mismatch on satisfaction.

Finally, we compared actor and partner effects in LL mismatch–satisfaction associations (see Table 4). The bivariate associations suggested both effects. For example, for men, relationship satisfaction was related both to their own and the woman's mismatch ($r$s of -.40 and -.36). To conduct a more rigorous comparison, we ran regression analyses with both men and women LL mismatch as predictors, by gender and by satisfaction scale. At the first step, these analyses controlled for length of relationship, which may be a confound of satisfaction. The second step entered either men or women LL mismatch, and the final step included both mismatch variables. Such a procedure allowed to examine the added value of each step of the model and to test for the incremental validity of actor vs. partner effects. For all analyses, both men and women LL mismatch contributed significantly to the regressions when entered separately at Step 2, consistent with the bivariate results. For men, the final regressions at Step 3 showed independent effects of both mismatch variables; male relationship and sexual

**Table 4. Regression models predicting male and female relationship satisfaction and sexual satisfaction with both partners' mismatching on LLs.**

| | B | SE | β | p | F | $R^2(\Delta R^2)$ |
|---|---|---|---|---|---|---|
| **Model 1. Dependent variable: Men's relationship satisfaction** | | | | | | |
| | B | SE | β | p | F | $R^2(\Delta R^2)$ |
| *Step 1* | | | | | | |
| Relationship length | −.00 | .00 | −.23 | .02 | 5.54 | .05 |
| *Step 2a* | | | | | | |
| Relationship length | −.00 | .00 | −.23 | .02 | 10.70 | .18 |
| Men's mismatching on LL | −.12 | .03 | −.36 | < .01 | | (.13) |
| *Step 2b* | | | | | | |
| Relationship length | −.00 | .00 | −.23 | .01 | 12.89 | .21 |
| Women's mismatching on LL | −.12 | .03 | −.40 | < .01 | | (.16) |
| *Step 3* | | | | | | |
| Relationship length | −.00 | .00 | −.22 | .00 | 10.28 | .24 |
| Men's mismatching on LL | −.07 | .03 | −.21 | .04 | | (.06[†]) |
| Women's mismatching on LL | −.09 | .03 | −.29 | < .01 | | |
| **Model 2. Dependent variable: Men's sexual satisfaction** | | | | | | |
| | B | SE | β | p | F | $R^2(\Delta R^2)$ |
| *Step 1* | | | | | | |
| Relationship length | .00 | .00 | −.11 | .30 | 1.11 | .01 |
| *Step 2a* | | | | | | |
| Relationship length | .00 | .00 | −.10 | .29 | 8.39 | .15 |
| Men's mismatching on LL | −.15 | .04 | −.37 | < .01 | | (.14) |
| *Step 2b* | | | | | | |
| Relationship length | −.00 | .00 | −.10 | .29 | 10.15 | .17 |
| Women's mismatching on LL | −.15 | .04 | −.40 | < .01 | | (.16) |
| *Step 3* | | | | | | |
| Relationship length | .00 | .00 | −.10 | .29 | 8.54 | .21 |
| Men's mismatching on LL | −.09 | .04 | −.22 | .04 | | (.06[†]) |
| Women's mismatching on LL | −.11 | .04 | −.29 | < .01 | | |
| **Model 3. Dependent variable: Women's relationship satisfaction** | | | | | | |
| | B | SE | β | p | F | $R^2(\Delta R^2)$ |
| *Step 1* | | | | | | |
| Relationship length | −.01 | .00 | −.37 | < .01 | 15.12 | .13 |
| *Step 2a* | | | | | | |
| Relationship length | .00 | .00 | −.36 | < .01 | 13.77 | .22 |
| Women's mismatching on LL | −.09 | .03 | −.30 | < .01 | | (.09) |
| *Step 2b* | | | | | | |
| Relationship length | −.01 | .00 | −.36 | < .01 | 14.36 | .23 |
| Men's mismatching on LL | −.11 | .03 | −.31 | < .01 | | (.02) |
| *Step 3* | | | | | | |
| Relationship length | −.00 | .00 | −.36 | < .01 | 10.92 | .25 |
| Women's mismatching on LL | −.06 | .03 | −.19 | .07 | | (.03[†]) |
| Men's mismatching on LL | −.73 | .35 | −.21 | .04 | | |
| **Model 4. Dependent variable: Women's sexual satisfaction** | | | | | | |
| | B | SE | β | p | F | $R^2(\Delta R^2)$ |
| *Step 1* | | | | | | |
| Relationship length | .00 | .00 | −.16 | .11 | 2.54 | .03 |
| *Step 2a* | | | | | | |
| Relationship length | .00 | .00 | −.16 | .12 | 3.53 | .07 |

*(Continued)*

**Table 4.** (Continued)

| | | | | | | |
|---|---|---|---|---|---|---|
| Women's mismatching on LL | −.08 | .04 | −.21 | .04 | | (.04) |
| *Step 2b* | | | | | | |
| Relationship length | .00 | .00 | −.15 | .11 | 4.51 | .09 |
| Men's mismatching on LL | −.10 | .04 | −.24 | .01 | | (.06) |
| *Step 3* | | | | | | |
| Relationship length | .00 | .00 | −.15 | .00 | 3.33 | .10 |
| Women's mismatching on LL | −.04 | .04 | −.11 | .32 | | (.03†) |
| Men's mismatching on LL | −.08 | .05 | −.19 | .10 | | |

*Note. Mismatching on LL* = the degree of discrepancy between one's preferred and partners' expressed LL; lower values indicate a better match. RS = Relationship satisfaction, SS = Sexual Satisfaction.

†Compared with step 2a.

satisfaction depends on both actor and partner effects. For women, the final regression for relationship satisfaction was similar, but the female mismatch predictor fell just short of significance. Neither predictor was significant at Step 3 for sexual satisfaction. Women's sexual satisfaction was more weakly predicted overall than the other outcome measures in these analyses (10% variance explained vs. 21–25%).

Overall, both actor and partner effects were found, providing further support for both the first and second hypotheses. We also tested for interactive effects of both partners' LL matching indicators in all four analyses. No significant interaction effects were found, indicating that benefits of LL matching are additive.

In addition, we found that relationship length was more consistently associated with lower relationship satisfaction than with sexual satisfaction. However, no significant correlation between LL mismatch and relationship length was observed for both men ($r = .01$, $p = .90$) and women ($r = .01$, $p = .91$), which suggests that partners do not adjust their expressed LL to their counterpart's preferences over time.

## Discussion

The present study sought to test empirically Chapman's [1] hypothesis of LLs and their association with relationship and sexual satisfaction, as well as to explore whether matching on LLs is associated with empathy. This was accomplished by examining partners' preferences for receiving and expressing love and assessing their relationship satisfaction, sexual satisfaction and four features of empathy.

Our study provides novel evidence in support of Chapman's [1] notion that speaking one's partner love language leads to higher quality relationships and create a positive emotional climate within the couple. In particular, the findings supported our major hypothesis that individuals whose partners express love in the way they prefer to receive it experience elevated relationship and sexual satisfaction. Previous work on sexual satisfaction has focused on discrepancies in desire as an influence on dissatisfaction [e.g. 43]. The present data suggest a broader role for discrepancy in displays of love that are not overtly sexual. The substantial positive correlations found between relationship and sexual satisfaction are consistent with previous research [44].

There were no significant gender differences in relationship and sexual satisfaction, nor in overall LL mismatch. However, women scored higher than men on four of the five "feeling" scales, indicating greater levels of need than men, with the largest effect sizes found for the desires for quality time and words of affirmation. This result contrasts with previous findings

that men appear to rely more on their partner for social and emotional support than women do [45, 46]. Possibly, men are more focused on fulfilling the social role of being in a committed relationship than specific affectionate behaviors, whereas women require more visible signs of love from their partner. Dykstra and Fokkema [46] suggest that women are more strongly oriented towards expressive and nurturing functions in marriage, implying greater awareness of whether their male partners is providing affirmations of love. Socialization to be emotionally independent may also contribute to men downplaying their needs for intimacy and affection [47]. Thus, the LL mismatch variable may not fully capture gender differences in the experience of relationships. Despite differences in the means of the "feeling" variables, our main predictions were supported for both male and female participants, implying that LLs function similarly in both genders.

In addition, findings supported the second hypothesis, that the actor's satisfaction would be negatively associated with the partner's LL mismatch. The regression analyses confirmed that both actor and LL mismatch predicted lower satisfaction. In the analyses of male respondents, both types of mismatches added significantly to the variance explained; findings in women showed a similar trend although not all effects reached significance, especially for sexual satisfaction. Partner effects tended to be slightly stronger predictors of relationship satisfaction and sexual satisfaction than the person's own degree of matching. Previous findings in the field of positive psychology [e.g. 17, 18] showing that acts of giving such as prosocial spending may potentially be more satisfying and lead to greater happiness, when compared to benefiting oneself. The partner effect is important in that it implies that LL matching is related to the actual quality of loving actions and communication between partners, and not merely individual perceptions. By contrast, higher emotional intelligence benefits the actor but not the partner, implying that this factor enhances actor perceptions of relationship quality, but it is not transmitted to the partner so that the benefit is internal to the actor [48].

Our findings were also in line with the previous research conducted with respect to other forms of behaviors that contribute to the quality of relationships. For instance, in two studies, relational maintenance strategies strongly predicted commitment, relational satisfaction, stability and loving others [49, 50]. Among other maintenance strategies, the studies examined positivity (e.g., upbeat during conversations, avoiding criticism), assurance (e.g., expressions of love, affirming commitment), social networks (e.g., spending time with common friends) and sharing tasks (e.g., engaging in household chores), which also resemble some aspects of the Words of Affirmation, Quality Time or Acts of service LLs. In addition, romantic physical touch was previously found to be strongly correlated with the relationship and partner satisfaction [51]. Receiving gifts was yet another factor that was previously found to be positively associated with the relationship strength, perceived similarity, as well as evaluation of the relationships' future potential [e.g. 52, 53].

The regression analyses showed that LL–satisfaction associations remained robust with length of relationship controlled. Relationship length was negatively correlated with the relationship satisfaction, as in other studies [e.g. 54, 55]. However, no meaningful association was observed between the length of relationship and matching on LLs, implying that people do not necessarily learn LLs of their partners with time. A focus on LLs might thus be of value in relationship counseling.

To our knowledge, this is the first study that found empirical support for LL hypothesis, contrasting with Egbert and Polk's study [31], which found no significant association between matching on LLs and relationship satisfaction. This discrepancy in the results between the previous research and our findings may a consequence of the method used to assess LLs. In our analysis we treated felt and expressed LLs as continuous dimensional qualities rather than categorizing participants based on a single dominant LL. The method based on collapsing couple

types (e.g. matched, partially matched, or mismatched) appeared to exaggerate differences in preferences for the five LLs. Dimensional assessment may be preferable to typologizing LLs. In the light of the present data, Chapman's [1] claims that people tend to manifest one, dominant LL should be revised in favor of assessing multiple preferences for expressing and giving love using the five LLs.

We further hypothesized that empathy may be a factor that drives matching on LLs, which in turn leads to the higher relationship satisfaction and sexual satisfaction among romantic partners. We aimed to check whether people, who scored higher on empathy scale (perspective taking and empathetic concern subscales, in particular), would also be more successful in expressing love in the way their partners prefer to receive it. Nevertheless, we did not find support for this prediction. Although small significant associations between matching in LLs and some empathy subscales (namely: perspective taking and fantasy) were observed among men, analogical effects were not observed in women. It is possible that because men tend to be less empathetic than women [e.g. 56, 57], the effect that high scores on empathy have on their relationships is stronger. However, the two significant associations in men did not transfer into significantly higher satisfaction among female partners.

Thus, benefits of LL matching on relationship and sexual satisfaction do not appear to be mediated by empathy. From a relationship counseling perspective, the implication is that training couples to improve matching may be addressed instrumentally as a skill to be acquired. For instance, such an intervention program could begin with measuring the level of discrepancies between partners in expressed vs. felt LLs, and providing both partners with ideas for specific behaviors that they could use to better meet the emotional needs of each other, as endorsed by Chapman [1]. Expressing love in the form desired may enhance relationship quality, even if the actor lacks cognitive and emotional insight into the partner's needs.

There are several other strengths and limitations of the current study that should be noted. Our sample included participants from different age groups, cultural backgrounds and with various relationship duration, which might serve as an indicator of greater generalizability and external validity of the findings. On the other hand, the multicultural sample might have been a confounding variable, which has influenced the findings in an uncontrolled way. Future research could study bigger and more homogeneous samples. Although in the precent study we have controlled for relationship length, when testing our hypotheses, in the subsequent researches more could be done to examine the association between the relationship length and matching on LLs (e.g., controlling for log or inverse values of the relationship duration).

This study applied an online self-report survey method which results in several significant limitations as well as response biases related to social desirability [58]. Other potentially problematic issues associated with the present study design include increased possibility of condescending responses, participants' self-selection, low response rate, submitting multiple responses, duration of the questionnaire, biased distribution channels as well as limited introspective ability of the respondent. At the same time, these methods could be associated with greater self-disclosure due to an online disinhibition effect [59].

The lack of suitable remuneration for the participants may have resulted in the response biases and nonprobability sampling. Other methods of assessment, as well as strategies aimed to prevent the response biases (i.e. adding several attention checks) and encouraging subjects providing reimbursement for participation are needed to eliminate the response and participation biases.

Another point that has to be addressed is the direction of causality. Even though the present results confirmed LL–satisfaction associations, future research should seek to establish the direction of causation between those factors. For instance, there is a possibility that romantic partners who are more satisfied in their current relationships are also more likely to give and

appreciate LLs behaviors. Our study also did not assess other factors that might drive matching on LLs. The observed association could also be multi-directional or be mediated by another variable. For instance, emotional intelligence may contribute to romantic relationship satisfaction and quality [see 60, 61]. Matching on LLs may also be a byproduct of assortative mating for personal characteristics like intelligence or personality type [62]. Other individual differences that were shown to influence relationship quality, such as time perspective or chronotype, could be also taken into account as potential mediators or confounders of the effects of LLs on relationship outcomes [e.g. 55; 63].

In conclusion, this study provides a unique contribution to the empirical literature on Chapman's basic assumption of Five Love Languages [1]. Our findings suggest that people who better match each other's preferences for LLs are more satisfied with their relationships and sexual life. Moreover, it appears that satisfying the needs of one's partner has at least as strong an impact on the individual's perceptions of relationship quality as receiving expression of love in the desired ways does. However, contrary to our hypothesis, small associations between matching on LLs and degree of empathy were observed only for some empathy subscales among male, but not female participants, and empathy was unrelated to satisfaction. Future work may explore other possible mediators of the LL matching–relationship satisfaction association.

The present findings, particularly the novel way to assess matching for LLs presented in the present paper, may be important for the subsequent research in the field of the romantic relationships. They also provide useful practical implications for marital and family counseling, as well as for laymen who aim to improve the quality of their relationship. Learning to recognize and react to one's partners love needs may be an important skill for building relationship satisfaction in both partners.

## Supporting information

**S1 Appendix.**
(DOCX)

**S1 Data.**
(CSV)

**S1 File. Variables' names and abbreviations.**
(DOCX)

## Author Contributions

**Conceptualization:** Olha Mostova.

**Data curation:** Olha Mostova.

**Formal analysis:** Olha Mostova, Maciej Stolarski.

**Investigation:** Olha Mostova.

**Methodology:** Olha Mostova, Maciej Stolarski.

**Project administration:** Maciej Stolarski.

**Supervision:** Maciej Stolarski.

**Writing – original draft:** Olha Mostova.

**Writing – review & editing:** Olha Mostova, Maciej Stolarski, Gerald Matthews.

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
