## [Decision Letter · Decision Letter 0]

4 Apr 2022

PONE-D-21-38702I love the way you love me: Responding to partner’s love language preferences boosts satisfaction in romantic heterosexual couplesPLOS ONE

Dear Dr. Stolarski,

Thank you for submitting your manuscript to PLOS ONE. After careful consideration, we feel that it has merit but does not fully meet PLOS ONE’s publication criteria as it currently stands. Therefore, we invite you to submit a revised version of the manuscript that addresses the points raised during the review process.

Please provide the following information/changes:

Methods: Questionnaires:. Provide information about the dates on which the questionnaire was applied on the average time spent on responses (discuss whether there could have been condescending responses and other response biases), and whether there were several attention checks. Clarify if there were payments to participants as a way to encourage participation.

In research in which self-response questionnaires are used, there could have been condescending responses and other response biases related to the social desirability.  The results of online surveys may be affected by bias due for example to low response rates, to a self-selection linked to the salience of a topic, the length of time required to complete the survey, the presentation of the questionnaire, the contact delivery modes, the use of pre-notifications and the presence of incentives. Furthermore, in online surveys, subjects often have greater self-disclosure. The results of the web-based surveys must consider all these aspects to be considered valid and reliable.  Moreover, about bias in participation, note the possible self-selection bias and the nonprobability sampling used to select the participants. See for example Van de Mortel, T.F. Faking it: Social desirability response bias in self-report research. Aust. J. Adv. Nurs. 2008, 25,40. Please provide information about this topic on methods section.

Did you have any missing data or did everyone answer all the questions? If you did have missing data, how did you treat the "missingness"?

We look forward to receiving your revised manuscript.

Kind regards,

Sonia Brito-Costa, Ph.D.

Academic Editor

PLOS ONE

Journal Requirements:

2. Please change "female” or "male" to "woman” or "man" as appropriate, when used as a noun (see for instance https://apastyle.apa.org/style-grammar-guidelines/bias-free-language/gender).

3. In the ethics statement in the Methods and online submission information, please clarify whether consent was written or verbal.  If verbal, please also specify: 1) whether the ethics committee approved the verbal consent procedure, 2) why written consent could not be obtained, and 3) how verbal consent was recorded. If your study included minors, state whether you obtained consent from parents or guardians. If the need for consent or parental consent was waived by the ethics committee, please include this information.

5. Your abstract cannot contain citations. Please only include citations in the body text of the manuscript, and ensure that they remain in ascending numerical order on first mention.

Reviewers' comments:

Reviewer's Responses to Questions

**Comments to the Author**

1. Is the manuscript technically sound, and do the data support the conclusions?

Reviewer #1: Yes

2. Has the statistical analysis been performed appropriately and rigorously? 

Reviewer #1: Yes

3. Have the authors made all data underlying the findings in their manuscript fully available?

Reviewer #1: Yes

4. Is the manuscript presented in an intelligible fashion and written in standard English?

Reviewer #1: Yes

5. Review Comments to the Author

Please provide the following information/changes:

Methods: Questionnaires:. Provide information about the dates on which the questionnaire was applied on the average time spent on responses (discuss whether there could have been condescending responses and other response biases), and whether there were several attention checks. Clarify if there were payments to participants as a way to encourage participation.

In research in which self-response questionnaires are used, there could have been condescending responses and other response biases related to the social desirability.  The results of online surveys may be affected by bias due for example to low response rates, to a self-selection linked to the salience of a topic, the length of time required to complete the survey, the presentation of the questionnaire, the contact delivery modes, the use of pre-notifications and the presence of incentives. Furthermore, in online surveys, subjects often have greater self-disclosure. The results of the web-based surveys must consider all these aspects to be considered valid and reliable.  Moreover, about bias in participation, note the possible self-selection bias and the nonprobability sampling used to select the participants. See for example Van de Mortel, T.F. Faking it: Social desirability response bias in self-report research. Aust. J. Adv. Nurs. 2008, 25,40. Please provide information about this topic on methods section.

Did you have any missing data or did everyone answer all the questions? If you did have missing data, how did you treat the "missingness"?

6. PLOS authors have the option to publish the peer review history of their article (what does this mean?). If published, this will include your full peer review and any attached files.

Reviewer #1: No

---

## [Author Response · Author response to Decision Letter 0]

17 May 2022

Thank you for your email dated April 5th, 2022 that addresses the points raised during the review process. We highly appreciate the Reviewer’s generous comments on the manuscript. 

We have carefully reviewed them and revised the manuscript to address the following concerns, as well as the Journal Requirements.

We have addressed the editorial points as follows:

ad.1: “Provide information about the dates on which the questionnaire was applied on the average time spent on responses (discuss whether there could have been condescending responses and other response biases), and whether there were several attention checks”.

The questionnaire was applied over the period between October 2018 to March 2019. The subjects spent an average of 23 minutes responding to the questionnaire. 

We have used the reverse wording items for each inventory to screen out careless respondents. No additional attention checks were used in the survey. 

We are aware that self-response questionnaires tend to be affected by the response biases. We have clarified it in the Methods section and proposed an additional future research suggestion to address the potential flaws.

The information is now provided in the manuscript in lines 236-239, and 562-569.

ad. 2: “Clarify if there were payments to participants as a way to encourage participation”.

No, the subjects participated in the survey voluntary and were not paid for their participation. It is now clarified in the Method section and addressed as a potential limitation in the Discussion.

The information is now provided in the manuscript in lines 237-238, and 570-574.

ad. 3: “In research in which self-response questionnaires are used, there could have been condescending responses and other response biases related to the social desirability…Please provide information about this topic on methods section”.

We have provided information on the potential biases in response and participation in our survey, including the possibility of the condescending responses. It is now addressed in the Method and Discussion sections.

The information is now provided in the manuscript in lines 562-574, and addressed in 238-244.

ad. 4: “Did you have any missing data or did everyone answer all the questions? If you did have missing data, how did you treat the "missingness"?”

Yes. Ten couples were discarded from the study due to significant amount of missing information for one or both partners. When one question was missed (i.e. not scored) by the respondent, the average over available items was calculated and multiplied with the number of items in the questionnaire to replace the missing value. 

The information is now provided in the manuscript in lines 215-222.

Formal suggestion: Please change "female” or "male" to "woman” or "man" as appropriate, when used as a noun

This point is now addressed across the entire manuscript; the only remaining uses of “male” and “female” are when these words are used as adjectives. 

We have attached a marked-up copy that highlights changes made to the original version, as well as unmarked version of our revised manuscript. If you have any further questions or suggestions, please do not hesitate to contact us directly.

---

## [Editor Report · Decision Letter 1]

23 May 2022

I love the way you love me: Responding to partner’s love language preferences boosts satisfaction in romantic heterosexual couples

PONE-D-21-38702R1

Dear Dr. Stolarski,

We’re pleased to inform you that your manuscript has been judged scientifically suitable for publication and will be formally accepted for publication once it meets all outstanding technical requirements.

Kind regards,

Sónia Brito-Costa, Ph.D.

Academic Editor

PLOS ONE
---

## [Editor Report · Acceptance letter]

27 May 2022

PONE-D-21-38702R1 

I love the way you love me: Responding to partner’s love language preferences boosts satisfaction in romantic heterosexual couples 

Dear Dr. Stolarski:

I'm pleased to inform you that your manuscript has been deemed suitable for publication in PLOS ONE. Congratulations! Your manuscript is now with our production department. 

Kind regards, 

on behalf of

Dr. Sónia Brito-Costa 

Academic Editor

PLOS ONE